# Virus–prokaryote infection pairs associated with prokaryotic production in a freshwater lake

Shang Shen,[1,2,3] Kento Tominaga,[4] Kenji Tsuchiya,[5] Tomonari Matsuda,[1] Takashi Yoshida,[6] Yoshihisa Shimizu[1]

**ABSTRACT** Viruses infect and kill prokaryotic populations in a density- or frequency-dependent manner and affect carbon cycling. However, the effects of the stratification transition, including the stratified and de-stratified periods, on the changes in prokaryotic and viral communities and their interactions remain unclear. We conducted a monthly survey of the surface and deep layers of a large and deep freshwater lake (Lake Biwa, Japan) for a year and analyzed the prokaryotic production and prokaryotic and viral community composition. Our analysis revealed that, in the surface layer, 19 prokaryotic species, accounting for approximately 40% of the total prokaryotic abundance, could potentially contribute to the majority of prokaryotic production, which is the highest during the summer and is suppressed by viruses. This suggests that a small fraction of prokaryotes and phages were the key infection pairs during the peak period of prokaryotic activity in the freshwater lake. We also found that approximately 50% of the dominant prokaryotic and viral species in the deep layer were present throughout the study period. This suggests that the "kill the winner" model could explain the viral impact on prokaryotes in the surface layer, but other dynamics may be at play in the deep layer. Furthermore, we found that annual vertical mixing could result in a similar rate of community change between the surface and deep layers. These findings may be valuable in understanding how communities and the interaction among them change when freshwater lake stratification is affected by global warming in the future.

**IMPORTANCE** Viral infection associated with prokaryotic production occurs in a density- or frequency-dependent manner and regulates the prokaryotic community. Stratification transition and annual vertical mixing in freshwater lakes are known to affect the prokaryotic community and the interaction between prokaryotes and viruses. By pairing measurements of virome analysis and prokaryotic production of a 1-year survey of the depths of surface and deep layers, we revealed (i) the prokaryotic infection pairs associated with prokaryotic production and (ii) the reset in prokaryotic and viral communities through annual vertical mixing in a freshwater lake. Our results provide a basis for future work into changes in stratification that may impact the biogeochemical cycling in freshwater lakes.

**KEYWORDS** viruses, prokaryotes, freshwater lakes, prokaryotic production, infection pairs, vertical mixing

Address correspondence to Shang Shen, s-shin@fc.ritsumei.ac.jp.

The authors declare no conflict of interest.

See the funding table on p. 15.

Viral infections are critical factors in prokaryote mortality because they suppress the dominance of prokaryotic species and resupply carbon and nutrients from prokaryotic cells to dissolved organic matter pools (viral shunts) (1, 2). Viruses attenuate 10%–60% of the daily prokaryotic production via infections (3, 4). Viral infection and host lysis can affect carbon processes, such as the microbial loop (5) and carbon pump (6, 7) which starts with prokaryotic production. Thus, identifying the infection pairs associated with prokaryotic production is a critical step in understanding the microbial

food web in aquatic environments. Recently, molecular ecology techniques have been developed to reveal the diversity of prokaryotes and viruses (8, 9), and the key interaction between viruses and prokaryotes that affects carbon cycling in various aquatic environments (9–12). The propagation and impact of viral infections depend on the abundance and activity of their hosts, and this relationship often follows a density- and frequency-dependent pattern, as described in the kill-the-winner model (13, 14). While this model is useful in describing free viruses (i.e., virions), it is relevant to the interactions between viruses and prokaryotes, which can resemble a predator–prey dynamic. A recent study has demonstrated the effectiveness of using viral metagenomes and co-occurrence network analysis to identify viruses that infect abundant prokaryotes (e.g., SAR11 and its phages) in Osaka Bay, Japan (15). Over a 2-year period, monthly surveys revealed clear patterns of co-occurrence and potential predation in these ecosystems (15). However, the key infection pairs associated with prokaryotic production remain unknown.

In freshwater lakes, unlike in oceans, the surface layer experiences dynamic changes in environmental parameters, such as water temperature, while the deep layer remains stable. This vertical mixing occurs once or twice a year, potentially affecting the composition of prokaryotic and viral communities, as well as their interactions during the shift from stratified to de-stratified periods. Previous studies have shown seasonal variation in the viruses that infect major prokaryotic lineages in both the surface layer (e.g., the phyla Actinobacteria, Cyanobacteria, and Bacteroidetes) and deeper layer (e.g., the phylum Planctomycetes) (9, 16–18). To date, only a few environmental studies have reported the depth profiles and seasonal variation of viral communities in freshwater lakes (9, 16, 17, 19). However, to date, no studies have conducted monthly surveys targeting both the surface and deep layers over the course of a year while analyzing prokaryotic production or revealed how the interactions between viruses and prokaryotes change with variations in the stratification of the water column.

Here, we describe how stratification transition affects communities and prokaryotic production in the surface and deep layers and which infection pairs can contribute to prokaryotic production. To achieve this, we collected lake water samples from both the surface and deep layers in Lake Biwa, a freshwater lake in Japan, over the course of a year, and analyzed the seasonal variation in prokaryotic production, and prokaryotic and viral communities. In the summer, higher prokaryotic production leads to higher viral infection in the surface layer, and approximately 70% of the production is reconnected to the dissolved organic matter pool via a viral shunt (20–22). Thus, we used an *in silico* host prediction analysis to identify virus–host infection pairs (23) and attempted to extract the key infection pairs that contribute to prokaryotic production using a co-occurrence network analysis among the predicted infection pairs.

## RESULTS

### Water temperature and dissolved oxygen concentration

Water temperature in the surface layer decreased from 24.1°C in September 2018 to 8.5°C in February 2019 and increased again to 28.9°C in August 2019, which was the highest in this study period (Fig. 1A). In the deep layer, the water temperature varied between 7.8°C and 9.4°C, which was lower and less variable than that in the surface layer. The de-stratified period, during which the water column was vertically and completely mixed, was defined as the period when the water temperature was the same in the surface and deep layer (<1°C), and the dissolved oxygen (DO) was supplied to the deep layer. In the present study, we considered that February and March represented the de-stratified period (10.4–10.9 $mgO_2$/L in the surface layer and 10.1–10.3 $mgO_2$/L in the deep layer, Fig. 1B), whereas the other months represented the stratified period. We divided the stratified period into the following three parts (Table 1): the beginning of the stratified period (April to June 2019), the middle of the stratified period (September 2018 and July to September 2019), and the end of the stratified period (October 2018 to January 2019 and October to December 2019). Notably, during this study period,

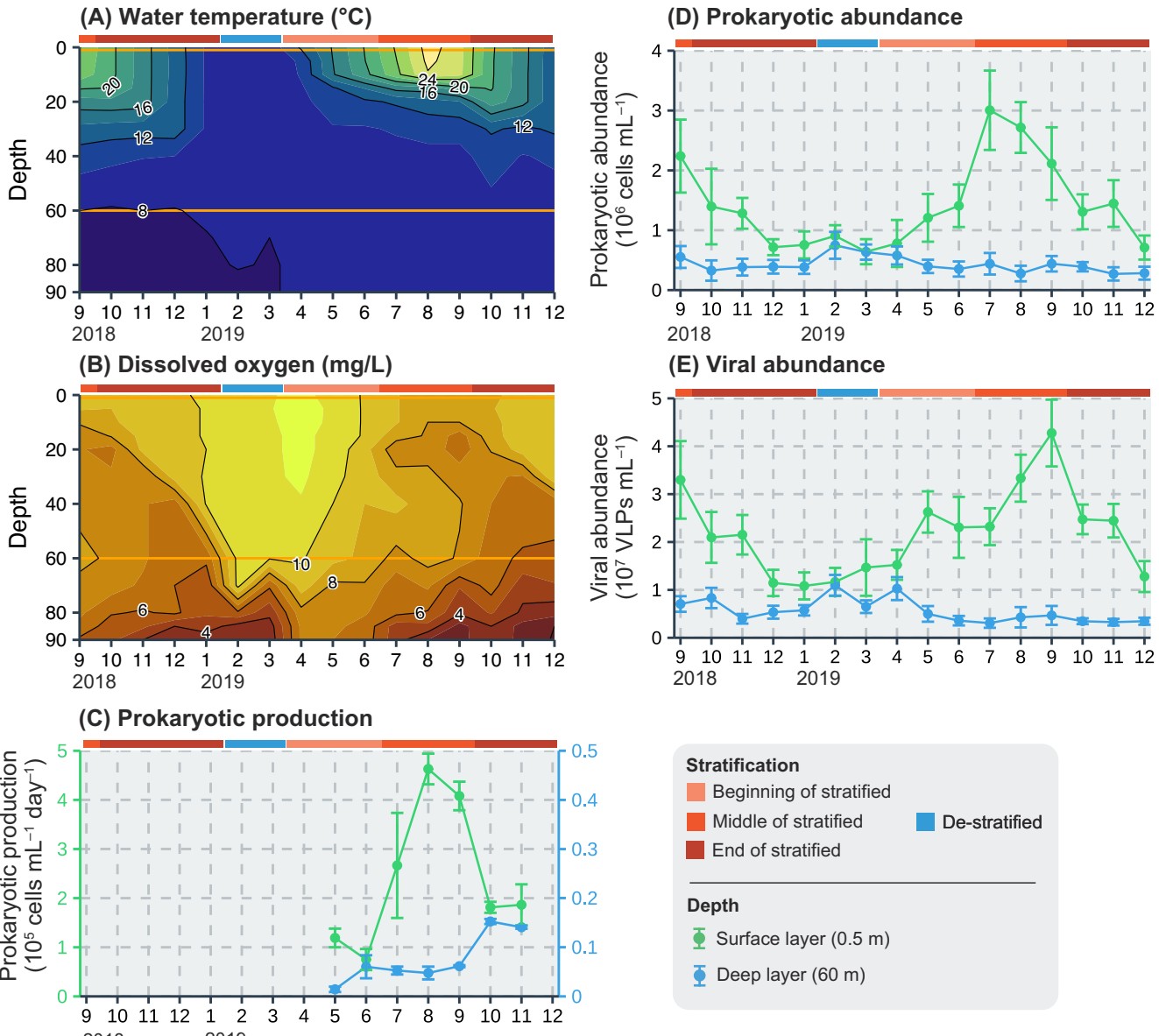

**FIG 1** Seasonal variation in the environmental parameters through the study period: (A) water temperature (°C), (B) concentration of dissolved oxygen (mg/L), (C) prokaryotic production (cells mL$^{-1}$ d$^{-1}$), (D) prokaryotic abundance (cells mL$^{-1}$), and (E) viral abundance (virus-like particles (VLPs) mL$^{-1}$). Orange lines in panels (A) and (B) indicate the sampling depths (0.5 and 60 m). Values in panels (C), (D), and (E) represent the mean ± standard deviation [$n = 3$ in panel (C) and $n = 10$ in panel (D, E)]. It should be noted that prokaryotic production was measured from May (at the onset of water column stratification) to November 2019 (when stratification was weakened). The Student's and Welch's $t$-tests were applied to test the significance of differences in prokaryotic and viral abundance, and prokaryotic production between the surface and deep layers. Results of the statistical analyses are shown in Table S1.

incomplete vertical mixing was first observed during winter (January to March) since regular monitoring began in Lake Biwa in 1979 (24). However, in February 2019, DO at a high concentration (> 10 mg/L) was supplied to a depth of 60 m, which was the sampling depth used in this study. This indicated that the water column from the surface layer to the depth of 60 m was completely mixed (Fig. 1B). Thus, the de-stratified period (February and March 2019) could be defined in this study and the prokaryotic/viral community between de-stratified and stratified period could be compared.

**TABLE 1** Basic information on season classification

| | | 2018 | | | | 2019 | | | | | | | | | | | |
|---|---|---|---|---|---|---|---|---|---|---|---|---|---|---|---|---|---|
| | | 9 | 10 | 11 | 12 | 1 | 2 | 3 | 4 | 5 | 6 | 7 | 8 | 9 | 10 | 11 | 12 |
| Stratified | Beginning | | | | | | | | + | + | + | | | | | | |
| | Middle | + | | | | | | | | | | + | + | + | | | |
| | End | | + | + | + | + | | | | | | | | | + | + | + |
| De-stratified | | | | | | | + | + | | | | | | | | | |

## Prokaryotic production and prokaryotic and viral abundance

Prokaryotic production in the surface layer increased with the water temperature and reached $4.6 \times 10^5$ cells mL$^{-1}$ d$^{-1}$ in the middle of the stratified period (August 2019, Fig. 1C). Prokaryotic production in the deep layer varied at $0.014–0.15 \times 10^5$ cells mL$^{-1}$ d$^{-1}$ (median: $0.06 \times 10^5$ cells mL$^{-1}$ d$^{-1}$) and was significantly lower ($P < 0.05$) and less variable than that in the surface layer (Table S1). Prokaryotic production in the surface layer was 12- to 97-fold higher than that in the deep layer.

Prokaryotic and viral abundances in the surface layer (prokaryotes: $0.64–3.0 \times 10^6$ cells mL$^{-1}$, viruses: $1.1–4.3 \times 10^7$ virus-like particles (VLPs) mL$^{-1}$, values show minimum–maximum) were the highest during the middle of the stratified period, when the water temperature is also higher than that during the other seasons (Fig. 1D and E). These three parameters were less variable and significantly lower ($P < 0.001$) in the deep layer (prokaryotes: $0.27–0.75 \times 10^6$ cells mL$^{-1}$, viruses: $0.31–1.1 \times 10^7$ VLPs mL$^{-1}$, values show minimum–maximum) than in the surface layer. The prokaryotic and viral abundance in the surface and deep layers showed similar values when the de-stratified period began (i.e., February).

## Diversity and seasonal variation of prokaryotic communities

A total of 1,608 amplicon sequence variants (ASVs) were detected from 1,497,727 non-chimeric reads obtained from 32 samples using 16S rRNA gene amplicon sequencing (Table S2). Actinobacteria was the most predominant phylum in both the surface and deep layers throughout the study period (Fig. 2A). Bacteroidetes and Cyanobacteria were also predominant in the surface layer, whereas Chloroflexi and Verrucomicrobia were predominant in the deep layer. According to the Bray–Curtis dissimilarity comparison between the surface and deep layers, the two communities were most closely related during the de-stratified period (0.084 in February 2019) and most different during the middle of stratified period (0.90 in September 2019, Fig. 3A). The non-metric multidimensional scaling (NMDS) analysis showed that the community in the surface layer was more variable than that in the deep layer (Fig. S1A). According to the dissimilarity comparing two samples based on the number of months between the sample collections, the dissimilarity in the two layers showed a local maximum around 6 months later (0.75 and 0.48 in the surface and deep layer, respectively) and a local minimum 12 months later (0.47 and 0.40 in the surface and deep layer, respectively; Fig. 3B). The dissimilarities of the prokaryotic communities in the surface layer after 6 months were significantly higher ($P = 1.1 \times 10^{-5}$) than those in the deep layer, and the dissimilarities after 12 months in the two layers showed no significant difference (Table S3).

## Diversity and seasonal variation of viral communities

A total of 13,761 Lake Biwa viruses (LBVs) were detected in 134,496,816 high-quality paired-end reads from the 27 samples (Table S4). The 385 LBVs had complete (i.e., circular) genomes. A total of 2,780 LBVs successfully predicted their hosts at the phylum and class level [Table S5, 10.6%–29.5% of total LBVs transcripts per kilobase of contig per million mapped reads (TPM)-based abundance]. Of these LBVs, viruses specific to members from the phyla Actinobacteria and Cyanobacteria were predominant in the surface layer, whereas those specific to members from the phyla Actinobacteria,

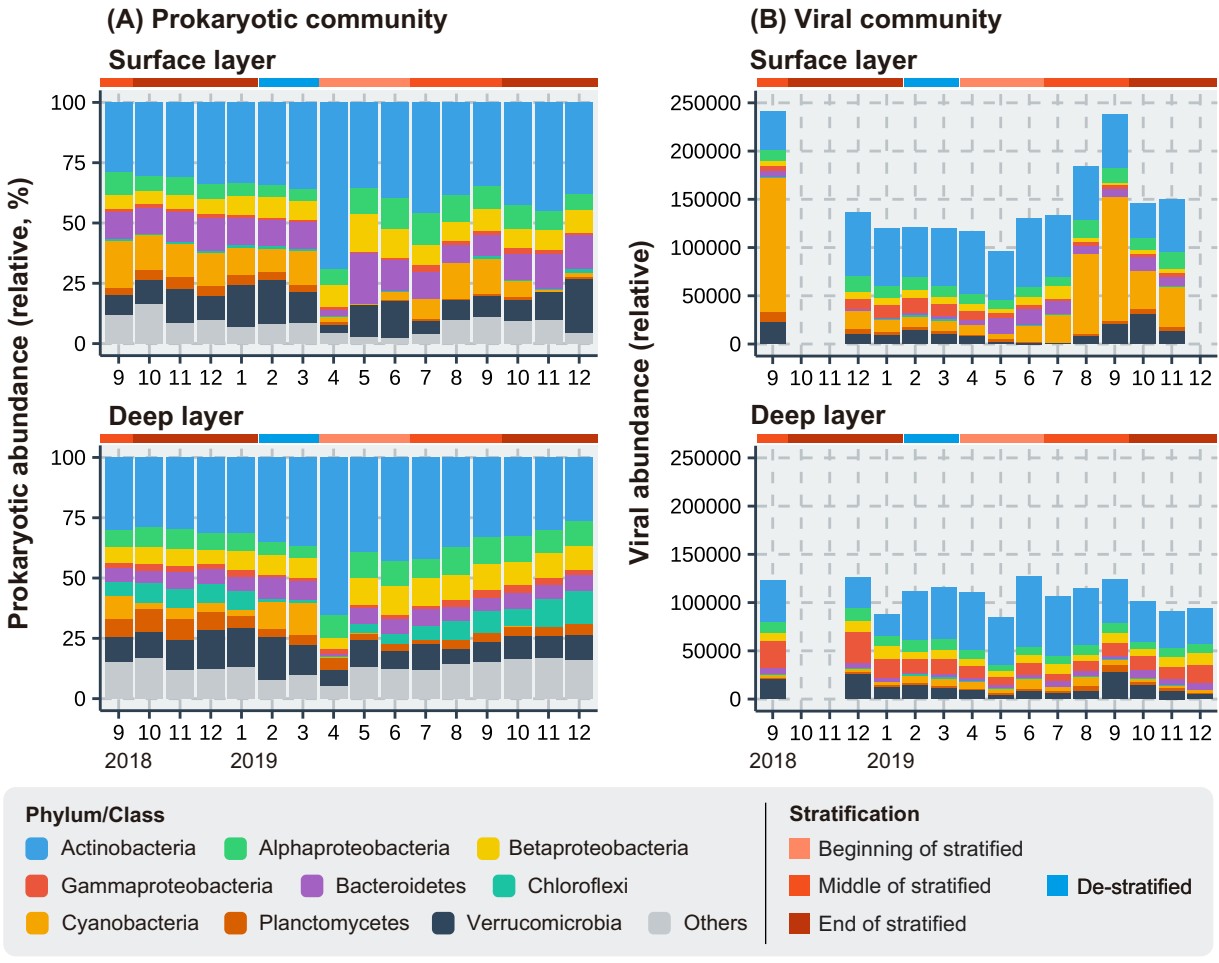

**FIG 2** Seasonal variation in the (A) prokaryotic community, shown as a percentage, and (B) viral community, shown as transcripts per kilobase per million, in the surface and deep layers during the study period. Viruses whose hosts were not predicted were excluded from the figures. Prokaryotes were classified by their phylum/class and viruses based on their predicted host phylum/class.

Gammaproteobacteria, and Verrucomicrobia were predominant in the deep layer (Fig. 2B). The diversity of the viral community (Shannon *H'*) was higher than that of the prokaryotic community (Fig. S2).

$P_{Habitat}$ in the de-stratified period showed a distribution with a center of 0.5, whereas the $P_{Habitat}$ in the middle of the stratified period showed a distribution with both ends (<0.05, >0.95; Fig. 3C).

According to the Bray–Curtis dissimilarity, between the surface and deep layers, the dissimilarity was the highest in the middle of the stratified period (0.86 in September 2018 and 2019) and the lowest in the de-stratified period (February 2019: 0.085; Fig. 3A). The NMDS analysis showed that the community in the surface layer was more variable than that in the deep layer (Fig. S1B). According to the dissimilarity comparing two samples based on the number of months between the sample collections, the dissimilarity in the surface layer showed a local maximum 6 months later (0.77) and a local minimum 12 months later (0.56), whereas in the deep layer, the dissimilarity monotonically increased throughout the study period (Fig. 3B). The dissimilarities of the viral communities in the surface layer after 6 months were significantly higher ($P = 1.1 \times 10^{-6}$) than those in the deep layer (Table S3). After 12 months, the dissimilarity showed similar values in the surface and deep layer (0.56 and 0.54, respectively).

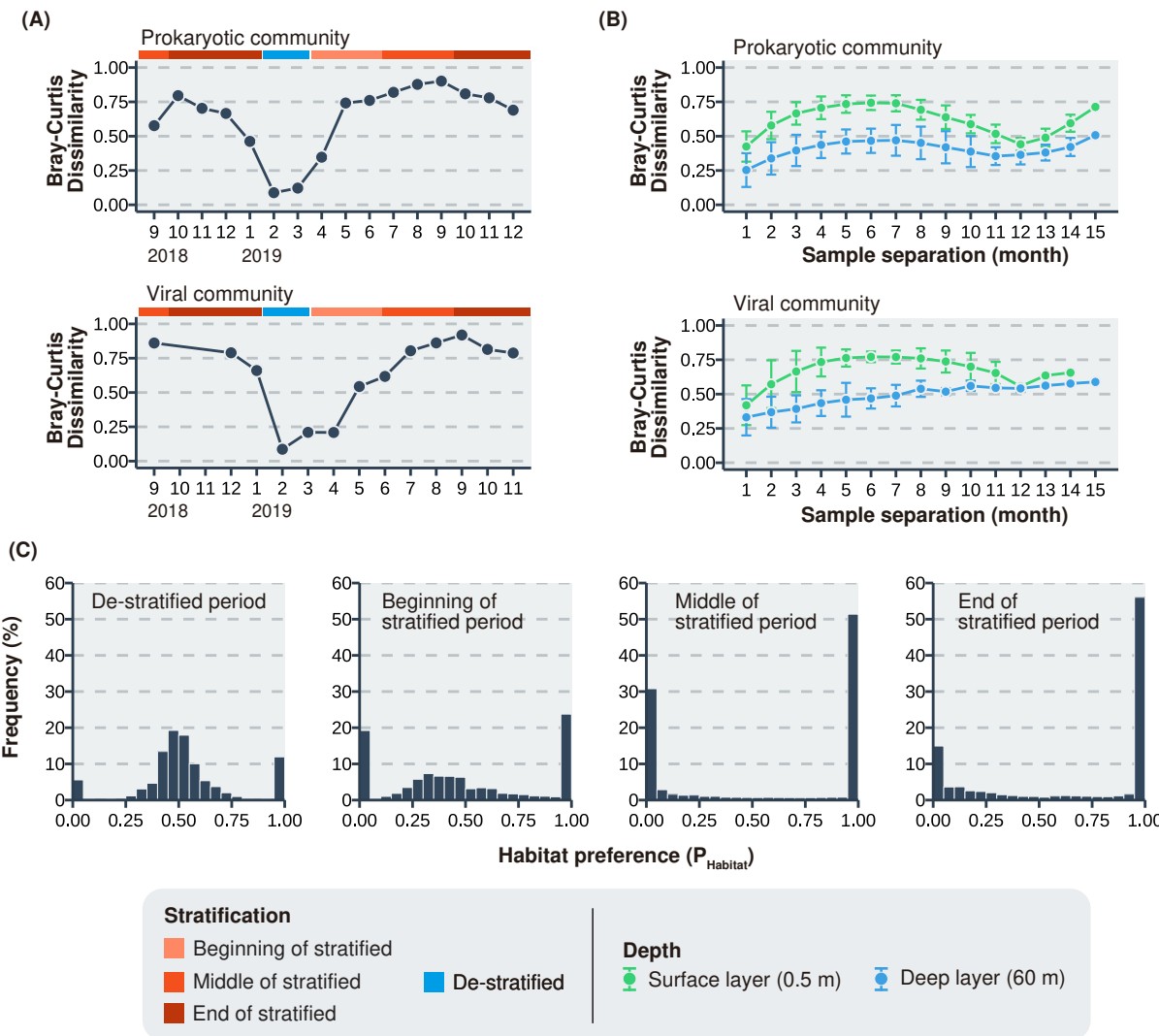

**FIG 3** Community analysis for prokaryotes and viruses. (A) Bray–Curtis dissimilarity of prokaryotic and viral community between the surface and deep layers. The sampling period was classified into four seasons, as described in Table 1. (B) Bray–Curtis dissimilarity of sample separation (month) for prokaryotic and viral communities. Each plot and error bar reflects the mean ± standard deviation. The Mann–Whitney U-test and Student's t-test were applied to test the significance of the observed differences in the dissimilarity of time-series separation. The results of the statistical analysis are shown in Table S3. (C) $P_{Habitat}$ of LBVs. LBV of $P_{Habitat}$ is 0, which means that LBV exists only in the deep layer. $P_{Habitat}$ indicates habitat preference, and LBVs indicate Lake Biwa viruses.

## Virus–prokaryote infection pairs related to prokaryotic production in the surface layer

A total of 21 ASVs ($ASV_{dominant}$, relative abundance is >1% during the middle of the stratified period), including those for Actinobacteria, Alphaproteobacteria, Betaproteobacteria, Bacteroidetes, Cyanobacteria, and Verrucomicrobia, showed the highest relative abundance in the surface layer (>1%) when the prokaryotic production was high (Fig. 4; Table S6A). The maximum relative abundance of these $ASV_{dominant}$ varieties varied from 1.0% to 14.1%. The total relative abundance of these 21 $ASV_{dominant}$ was 41.3% in July, 56.3% in August, and 46.2% in September. For 19 $ASV_{dominant}$ of the 21 $ASV_{dominant}$, at least one LBV co-occurred (Fig. S3; Table S6). Of the remaining two $ASV_{dominant}$, LBVs co-occurring with ASV_41 (phylum Chlorobi) could not be detected and ASV_204 (phylum Gemmatimonadetes) was excluded from the analysis because its phages could not be predicted. In total, 656 LBVs were found to co-occur with these 19 ASVs, accounting for approximately 6.9%, 18.9%, and 13.3% in July, August, and September, respectively, as determined based on the TPM-based relative abundance (Table S6B). Of

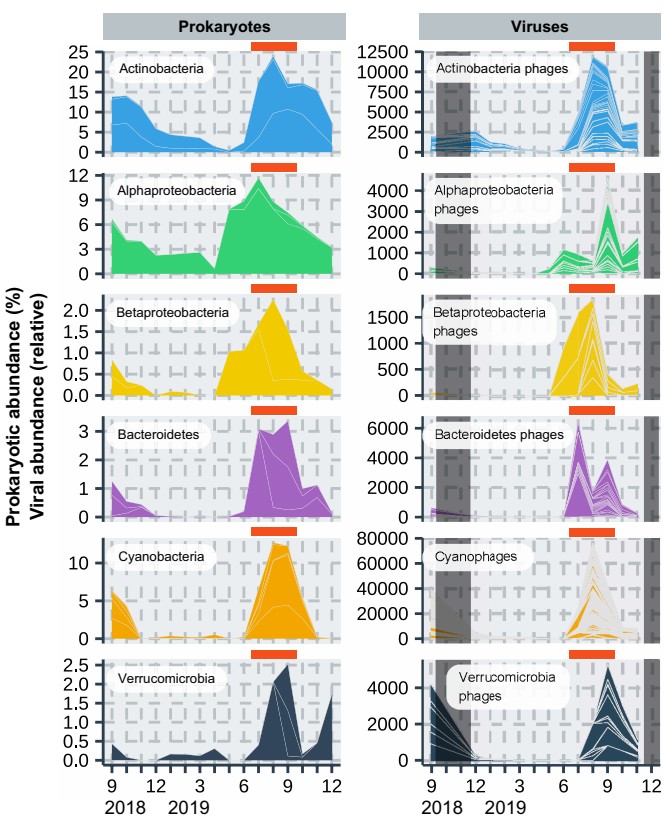

**FIG 4** Seasonal variation in the abundance of prokaryotic species (19 ASVs) may contribute to prokaryotic production and LBVs co-occurring with the ASVs in the surface layer. Gray shading indicates no data (Table S9). Individual plots for ASVs and co-occurring LBVs are shown in Fig. S3. ASVs indicate amplicon sequence variants, and LBVs indicate Lake Biwa viruses. Orange bars indicate the period from July to September in 2019 during which prokaryotic production was high.

the 19 $ASV_{dominant}$, 13 $ASV_{dominant}$ species were undetectable or showed low abundance before their abundance increased in the middle of the stratified period, and they consisted of Actinobacteria, Bacteroidetes, Cyanobacteria, Proteobacteria (alpha and beta), and Verrucomicrobia. The remaining six $ASV_{dominant}$ (ASV_4, 5, 7, 80, 84, 122) were also dominant in other periods.

## Temporal dynamics of dominant species in prokaryotic and viral community

The number of ASVs with relative abundances of more than 1% for at least 1 month during this study period was 99 and 65 in the surface and deep layers, respectively (Fig. 5A), corresponding to approximately 5% of the total ASVs (i.e., 1,,608 species) detected. Of these, in the surface layer, approximately 44 (40%) and 5 (5%) of the ASVs showed short (i.e., dominant for 1 month) and persistent (i.e., dominant for more than 12 months) patterns, respectively (Fig. 5A). Most of these existed (>0%) for less than 8 months (e.g., ASV_89, 169, 177, and 305 in Fig. S3). In the deep layer, approximately 23 (35%) and 15 (20%) of the ASVs showed short and persistent patterns, respectively. Furthermore, 50% of the dominant ASVs in the deep layer existed throughout the study period (16 months).

Dominant LBVs, those ranked within 100 based on the TPM data in the surface layer, showed a pattern similar to that of ASVs in the surface layer (Fig. 5B). Approximately 60% of LBVs showed a short pattern. In the deep layer, the dominant patterns of LBVs were different from those of ASVs. Approximately 40% of LBVs showed a short pattern but fewer LBVs showed a broad pattern. Moreover, 60% of dominant LBVs in the deep layer existed (TPM >0) throughout the study period, but only 20% of dominant LBVs in the surface layer existed throughout the study period.

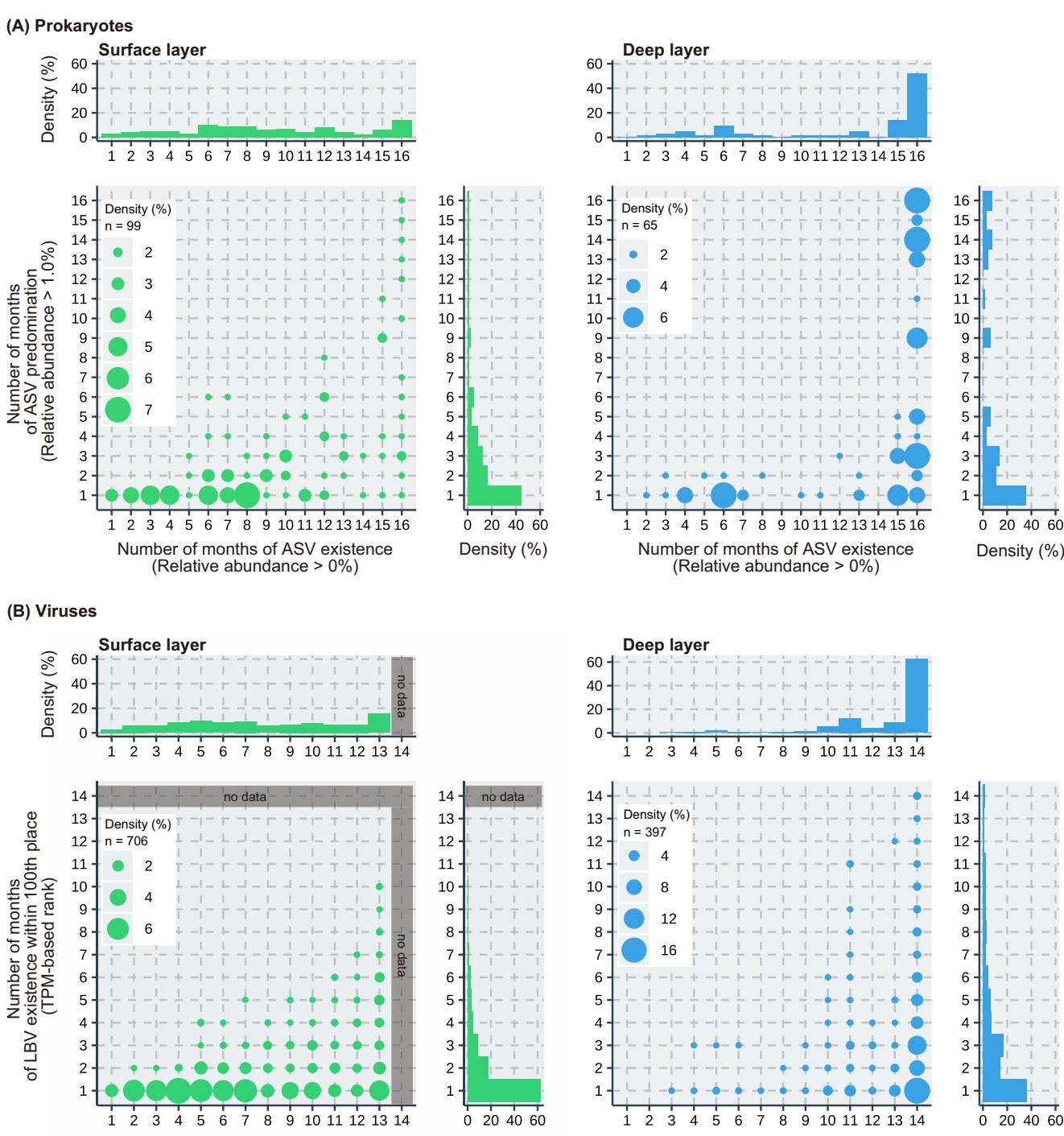

**FIG 5** Existence vs predominance plots for (A) prokaryotes and (B) viruses in the surface and deep layers. The x-axis indicates the number of months that ASV and LBV existed during the study period. The y-axis indicates the number of months for which ASV and LBV predominated. Predominant ASV is defined as that whose relative abundance is higher than 1%. Predominated LBV is defined as that with a TPM-based rank within 100 (13,761 LBVs were detected in this study).

## DISCUSSION

The data we obtained via annual sampling from the surface and deep layers of the freshwater lake led to two major findings. First, a small number of prokaryotes, accounting for approximately 50% of the abundance, may be infected and suppressed by viruses when prokaryotic production was high. Second, annual vertical mixing of the water

column may result in a similar rate of community change between the surface and deep layers.

## Prokaryotic species may exclusively drive prokaryotic production in summer and be infected by their phages in the surface layer

In the middle of the stratified period (Summer), prokaryotic production and prokaryotic and viral abundance were the highest in the surface layer throughout the study period (Fig. 1). The viral infection rate is also the highest in this period, and approximately 40z%–60% of prokaryotes are reconnected to the pool of dissolved organic matter via viral lysis (21, 22). One of the major successes of this study was that we estimated the potential virus–prokaryote infection pairs related to the prokaryotic production. Infection pairs, including 19 $ASV_{dominant}$ and 656 LBVs, were successfully estimated in the middle of the stratified period. These 19 $ASV_{dominant}$ accounted for 41.2%–52.7% of total prokaryotic abundance. Of the 19 $ASV_{dominant}$, 13 species were undetectable or showed low abundance in other periods but could grow more actively (r-strategy) than other ASVs in response to the increase in water temperature or dissolved organic matter derived from primary production at the middle of the stratified period (Fig. S3 and S5; Table S7). This is supported by the findings of previous studies (25–30). For instance, the growth rate and absolute abundance of *Synechococcus* (GpIIa) increased at warmer water temperatures (28, 29). Actinobacteria and Proteobacteria increased in their relative abundances with the increase in the labile substrate (27), and Verrucomicrobia could rapidly respond to carbohydrates or cyanobacterial bloom (25, 26, 30). Concentrations of these labile substances are higher in the surface layer during the middle of the stratified period in Lake Biwa (31). Thus, these 13 $ASV_{dominant}$ might be growing actively, thereby contributing to prokaryotic production, and can be infected by viruses based on the density- and frequency-dependent models (13, 14). In this study, bacteria that do not incorporate dA (lack of deoxyribonucleoside kinases) were not included in the prokaryotic production analysis (32). Moreover, minor (<1% of relative abundance) but active bacteria were not considered the $ASV_{dominant}$ in this study. However, a previous study using the bromodeoxyuridine-incorporating method reported that approximately 60% of prokaryotic cells showed growing activity in oligotrophic Lake Stechlin (Germany) (33), which is consistent with our result that the 19 $ASV_{dominant}$ accounted for 41.2%–52.7% of total prokaryotic abundance. These results supported that a few prokaryotic species contributed to most part of the prokaryotic production during the middle of the stratified period. Our results do not deny the possibility of grazing pressure on these 19 ASVs in the middle of the stratified period. Protists also actively graze prokaryotes in the middle of the stratified period in Lake Biwa (34), and these 19 ASVs may have also been grazed by protists. In this study, we only measured the prokaryotic production of free-living prokaryotes in the GF/C filtrates but not the prokaryotic production or interaction between prokaryotes and viruses attached to the particulate organic matter. Because viruses may affect the dynamics of particulate organic matter (35, 36), further studies should be conducted, such as determining which viral group may affect particle export.

In the deep layer, the number of ASVs with co-occurring viruses was lower than that in the surface layer (82 and 26 ASVs in the surface and deep layers, respectively). First, because the prokaryotic and viral community in the deep layer did not change dynamically, fewer infection pairs were detected using co-occurrence network analysis. Second, approximately 20% of LBVs could be predicted by their host phylum and class, and the remaining 80% remained unknown. LBVs in the deep layer (also in the surface layer) that were not detected to co-occur with ASVs could be included in the remaining LBVs. Third, oligotrophic environments (lower host abundance or prokaryotic production) favor lysogenic viruses, whereas lytic viruses are dominant in highly productive environments (37, 38). In the deep layer, prokaryotic production was markedly lesser than that in the surface layer (Fig. 1C), and the infection strategy of some viruses in the deep layer might be lysogeny. Furthermore, viruses in the deep layer could be lysogenic

when their host becomes abundant based on the Piggyback-the-Winner model (39). However, of the 397 LBVs that ranked within 100 based on the TPM data, only 14 LBVs carried the integrase gene (Table S8). To test this possibility, further analyses, such as analysis of the viral metagenome in prokaryotic cells, must be carried out (9). Fourth, this study only focused on dsDNA viruses. Recently, there has been a growing recognition of the importance of RNA and ssDNA viruses, which were removed during the purification step (CsCl density centrifugation, see Materials and Methods) in the oceans (40–43). As most of the described prokaryote-infecting viruses are dsDNA viruses (8), ssDNA and RNA viruses were excluded from the purview of the current study, which may limit the comprehensive understanding of viral diversity. Future studies should focus on a broader array of viral types by incorporating ssDNA viruses and RNA viruses into the analyses. Moreover, giant viruses that may be removed via filtration (pore size: 0.2 µm, see Materials and Methods) have been studied intensively in recent years (44, 45). However, giant viruses infecting prokaryotes have not been reported and were excluded from this research.

## Most prokaryotes and viruses in the deep layer persist during all months but dominate in only 1 month

In the surface layer, environmental parameters, such as water temperature and the phytoplankton community composition (40), change dynamically, resulting in changes in the composition of organic matter. This kind of rapid environmental change may favor approximately 40% of dominant ASV species, which supports the observed short patterns of abundant ASVs (relative abundance >1%) in the surface layer. Viruses also showed a pattern similar to that of ASVs in the surface layer, reflecting that these viruses increased in abundance by infecting and lysing their host prokaryotes.

In the deep layer, both patterns were observed for the dominant ASVs, and unlike the surface layer, more ASVs showed persistent patterns (i.e., >12 months). This can be explained by the environmental stability of deep layers. In the deep layer, water temperature is stable (7.8°C–9.4°C), and prokaryotic production is notably lower than that in the surface layer (Fig. 1A and C). Moreover, new organic production is limited (41) and usable (labile) organic matter is depleted (21), leading to the slow growth of prokaryotes. Notably, 23 ASVs in the deep layer showed a short pattern (Fig. 5A). Of these, seven ASVs increased during the de-stratified period and may have been supplied by the surface layer via vertical water mixing (e.g., ASV_11, 81, and 43 in Fig. S4). The other 16 ASVs (e.g., ASV_64 and 149 in Fig. S4) may increase in response to the substrate that may be provided by the surface layer through sinking particles or from lysates via infection of other ASVs. Interestingly, unlike the dominant ASVs, most of the dominant LBVs showed a short or intermediate (2–3 months) pattern, suggesting the result of ASVs with a short pattern. Moreover, most of these LBVs were present throughout the study period (i.e., 14 months). These results indicated that although the environment in the deep layer does not change considerably, some ASVs may occasionally become dominant and be suppressed by their phages, resulting in their phages ranking within 100. These dominant viruses in the deeper layer exist for a long time (10–14 months) probably because factors that remove viral particles—such as the particulate organic matter on which viruses attach or heterotrophic protists, which may ingest viral particles—are depleted. This could explain how ASV composition changed seasonally but LBV composition changed monotonically in the deep layer (Fig. 3B). Another possibility is that some dominant ASVs (>1%) are not infected by viruses. A previous study suggested that host species at an abundance of less than $10^4$ cells/mL are not effectively infected by their phages (42). The prokaryotic abundance in the deeper layer was less than $10^6$ cells/mL (Fig. 1D), and some dominant ASVs may not fulfill this effective infection threshold (i.e., $10^4$ cells/mL) even though their relative abundance was >1%. This may result in the viral community in the deeper layer changing monotonically.

## The rate of community changes is differentiated by stratification and is uniform annually owing to vertical mixing

The prokaryotic and viral communities between the surface and deep layers were substantially different in the middle of the stratified period and mixed during the de-stratified period (Fig. 3A). This may have been affected by the differences in community changes between the two layers. The higher dissimilarities of the prokaryotic and viral communities in the surface layer 6 months later than those in the deep layer indicated that the communities in the surface layer changed faster than those in the deep layer. The 6-month time gap leads to opposite seasons being considered (e.g., spring vs autumn, summer vs winter), and these differences in dissimilarity may result from differences in prokaryotic production or environmental parameters.

For the prokaryotic and viral communities, the dissimilarity between the surface and deep layers was not substantially different after 12 months (Fig. 3B; Table S3). Interestingly, the gap in community changes between the surface and deep layers closed after 12 months, although prokaryotic production and environmental parameters between the two layers were different. These results indicated that prokaryotic and viral communities in the surface and deep layers change at a similar rate over the years. This finding is inconsistent with that of previous research wherein the prokaryotic community in the surface layer (5 m) was shown to change faster than that in the deep layer (150 m) of the ocean (43, 44). Annual vertical mixing may explain why, unlike the ocean, community changes between the surface and deep layers were similar over years in Lake Biwa. Vertical mixing occurs each year (mostly from January to March) in Lake Biwa. In the de-stratified period, not only the abundance and composition of microorganisms but also environmental parameters (e.g., concentration of nutrients or organic matter) are vertically mixed and become similar throughout the water column (45, 46). Although rapid changes in prokaryotic production or environmental parameters in the surface layer result in faster changes in the prokaryotic and viral community, community changes are reset by vertical mixing in the de-stratified period every year. This could lead to a similar rate of change in prokaryotic and viral communities between the surface and deep layers after 12 months. Unlike in oceanic environments (43, 44), prokaryotic and viral communities in the deep layers of aquatic environments with vertical mixing may change at a rate similar to that in the surface layer.

Our monthly survey combining the community analysis and prokaryotic production revealed the way through which dominant prokaryotes are suppressed by a viral infection and identified the key infection pairs related to the prokaryotic production in the freshwater lake. The results of this study will help to understand how viruses affect carbon cycling via viral infections in prokaryotes that respond to primary production. Our analysis also indicated that the rate of community changes is similar between the surface and deep layers due to the annual vertical mixing even though the environments or prokaryotic activity in the surface and deep layer are different. This finding might aid in understanding the effect of climate change on stratification and the microbial community in the future.

## MATERIALS AND METHODS

### Sample and data collection

Thirty-two lake water samples were collected at the pelagic station in Lake Biwa (35°23′41″N, 136°07′57″E, total depth of the sampling station: 90 m) between September 2018 and December 2019 (Table S9), from two depth layers: surface layer (depth: 0.5 m) and deep layer (depth: 60 m).

A data set of environmental parameters (water temperature and DO concentration) was obtained from white papers published previously (45, 46).

## Analyses of prokaryotic production and prokaryotic and viral abundance

Prokaryotic production was measured using the [$^{15}$N5]–2-deoxyadenosine ($^{15}$N-dA)-incorporation method described earlier (20, 47). The incorporation rate of $^{15}$N-dA (pmol L$^{-1}$ d$^{-1}$) was converted to the bacterial growth rate (cell L$^{-1}$ d$^{-1}$) using a conversion factor [$1.83 \times 10^6$ cells (pmol $^{15}$N-dA)$^{-1}$] determined in the same lake (20). It should be noted that prokaryotic production was measured from May (at the onset of water column stratification) to November 2019 (when stratification was weakened) in order to encompass the period in which prokaryotic production showed substantial variability.

The collected lake water sample was fixed with buffered glutaraldehyde (pH 7.2, final concentration: 1%) immediately onboard, incubated at 4°C before transportation to the laboratory, and stored at −30°C. Slide glasses were prepared using DNA staining methods (48), except that the SYBR Gold kit (Thermo Fisher Scientific, Oregon, USA; stock solution diluted 1:400) was used instead of the SYBR Green I kits (49). Prokaryotic cells and virus-like particles were counted under an epifluorescence microscope (BZ-9000, KEYENCE) at 1,000× magnification. Ten fields with at least 200 cells or virus-like particles were counted per sample. The slides were stored at −30°C until counting.

## Partial 16S rRNA gene amplicon sequencing

The collected samples (250 mL) were prefiltered through GF/C filters (pore size: 1.2 µm, Merck) to remove bacteria attached to particulate organic matter and obtain free-living prokaryotes. The filtrates were filtered through polycarbonate filters (pore size: 0.2 µm). DNA was extracted using a DNeasy PowerWater Kit (Qiagen, Hilden, Germany). The V3-V4 region of the 16S rRNA gene was amplified using the primer pair 341F (CCTACGGGNGGCWGCAG) and 805R (GACTACHVGGGTATCTAATCC) (50). A sequencing library was prepared using the Illumina standard protocol (15044223 B, 16S Metagenomic Sequencing Library Preparation), and sequenced using a MiSeq sequencing system with a V3 reagent kit (300 × 2 bp, Illumina, San Diego, CA, USA).

The sequenced reads were demultiplexed and trimmed using Claident (v0.2.2019.05.10) by removing PCR primers and reads that included low-quality regions with default settings (51). To differentiate ASVs, the trimmed reads were analyzed using the DADA2 pipeline (v. 1.16) in R version 3.5.1, with the settings of "trimLeft = 17, trimRight = 21" (52, 53). ASVs were first classified using the ARB database (54) (55). The unclassified ASVs were classified using Claident with default settings against the National Center for Biotechnology Information (NCBI) nucleotide sequence database (51, 56). ASVs not assigned as prokaryotes (eukaryotes, mitochondria, and chloroplasts) were removed. The summary regarding the ASVs is presented in Table S10.

## Viral DNA preparation and sequencing

Four liters of collected lake water were prefiltered through GF/C filters (pore size: 1.2 µm) and then filtered through polycarbonate filters (pore size: 0.2 µm). The viral particles in the filtrates were concentrated using the Fe-based flocculation method (57). Although ssDNA and RNA viruses infect prokaryotes, most of the described prokaryote-infecting viruses are dsDNA viruses; therefore, we focused on this group (8). We extracted 1.5 g/mL of the CsCl layer to purify the viral particles (58). Free DNA was removed via DNase treatment (Recombinant DNase I; Takara, Japan). Viral DNA was extracted using the DNeasy Blood & Tissue Kits (Qiagen, Hilden, Germany). A sequencing library was prepared according to the Illumina standard protocol, including 12 cycles of the amplification step (15031942 Rev. C, Nextera XT DNA Sample Preparation Guide), except that we used 0.25 ng viral DNA as an initial input (59). The prepared library was sequenced using a MiSeq sequencing system with a V3 reagent kit (300× 2 bp; Illumina, four to six samples).

## Genome assembly and gene prediction and annotation

For the bioinformatics analysis, default parameters were used for all tools unless otherwise specified. Low-quality raw reads were removed using Trimmomatic v0.39 using default settings (60). The remaining raw reads were assembled using SPAdes v. 3.13.1 (61) using the "—careful" option with k-mer option lengths of 21, 33, 55, 77, 99, and 127 because SPAdes was identified as the best assembler for viromes according to a previous study [Table 1 in (59)]. All 27 samples were co-assembled using SPAdes using the same options. Only contigs longer than 10 kbp (29,730 contigs) were used for further analysis. VirSorter was used to remove non-viral contigs using the "--virome" option (62). Contigs classified as VirSorter category 1–3 ("most confident," "likely," and "possible" prediction (62)] were considered viral contigs and used for further analysis (Table S4, 24 967 contigs). The 84% of the assembled contigs (>10 kbp) were detected to be of viral origin by VirSorter (Table S4). These viral contigs were clustered using all-vs-all blastn at >95% nucleotide identity across >95% of their length to remove redundancy (16). The longest contig in each cluster was selected as the representative contig. The complete genome (i.e., circular genome) was determined using ccfind using default settings (59). Open reading frames (ORFs) were predicted using Prodigal v. 2.6.3 using the "-p meta" option (63). The gene function of the predicted open reading frames was annotated against multiple databases and tools for homology, domain, and family searches to predict more genes according to a previous study [see "Workflow for gene functional annotation" in the supplementary information (9)]. Briefly, the predicted ORFs were annotated using eggnog-Mapper (v. 1.0.3) with the "-m diamond" option and thresholds of $10^{-5}$ e-value (64, 65), using the DIAMOND (v. 0.9.29) against the UniRef90 (release 2018_01) database with thresholds of $10^{-5}$ e-value (66–68), and using the HMMER (v. 3.1b2) against the Prokaryotic Virus Orthologous Groups hidden Markov model (HMM) profiles with thresholds of $10^{-5}$ e-value (69, 70). ORFs were also searched against the Pfam v31.0 database (71) using a sensitive HMM-HMM search employing HHsearch and JackHMMER (72, 73) in pipeline_for_high_senstive_domain_search (https://github.com/yosuken/pipeline_for_high_sensitive_domain_search) (10, 59).

## Host prediction

LBV hosts were assigned using multiple methods (9, 23, 59). The priority of the prediction methods was set as method (i) > (ii) > (iii) > (iv) > (v) > (vi) when a different phylum/class was predicted in an LBV. (i) When a given LBV was highly similar ($S_G$ >0.15) to a reference viral genome whose host was reported, the host of the LBV was assigned to the phylum of the host of the reference virus (59). (ii) Actinobacteria and Cyanobacteria were identified as hosts of LBV because host-specific genes were found in a phage genome/contig [i.e., *whiB* for Actinobacteria (18) and a photosystem gene for Cyanobacteria (74)]. CRISPR spacers and tRNAs in prokaryotic genomes obtained from the NCBI database [search conditions: "(BCT) AND (WGS) AND ((Lake freshwater) OR (freshwater))"] and a previous study (9) were detected using metaCRT (75) and ARAGORN1.2.36 (76), respectively. Lake Biwa virus sequences were compared to (iii) the CRISPR spacers using BLASTn with a threshold of 100% identity and to (iv) the tRNAs using BLASTn with thresholds of >97% identity, alignment length ≥30 bp, and $<10^{-5}$ e-value. (v) Lake Biwa virus genomes were also directly compared with prokaryotic genomes using BLASTn with thresholds >97% identity, alignment length ≥30 bp, and $<10^{-5}$ e-value. The prokaryotic genomes used for this prediction (using CRISPR, tRNA, and direct comparison) were obtained from NCBI RefSeq (downloaded in June 2020) and Lake Biwa metagenome-assembled genomes (9). Other methods used for host identification were (vi) ">80% of the annotated bacterial genes were taxonomically related to a single taxon," as described earlier (9). The summary regarding the LBVs is presented in Table S11.

## Community data analysis

The relative abundance of LBVs was calculated as TPM using the CountMappedReads2 script (https://github.com/yosuken/CountMappedReads2). The coverage of each LBV in each sample was calculated using CoverM (https://github.com/wwood/CoverM). The TPM was set to zero when the coverage of the LBV was less than 80% of the genome/contig length (16). The habitat preference of each LBV in each season is calculated as follows (9):

$$P_{Habitat} = \frac{\sum TPM_{surface\ layer\ i}}{\sum TPM_{the\ surface\ layer,\ i} + \sum TPM_{deep\ layer,\ i}}$$

where i indicates months in certain seasons (e.g., i = 7, 8, and 9 in the middle of the stratified period, Table 1). $P_{Habitat}$ of a LBV = 0 indicates that the LBV exists only in the deep layer.

Before community analysis, the data set of each sample was rarefied based on coverage. The rarefied data sets were then used to calculate the Shannon index ($H'$) and generate NMDS plots. The dissimilarity of sample pairs (surface vs deep layer, two samples with monthly separation) was calculated based on the Bray–Curtis dissimilarity. These community analyses were performed using the vegan v.2.5–7 package (https://CRAN.R-project.org/package=vegan) in R language (52).

For the dominance patterns of prokaryotic species (ASVs), the following three categories were defined: the persistent patterns were defined as ASVs dominated during >75% of the study period, the short pattern was defined as ASVs dominated during <10% of the study period (but at least 1 month) as described earlier (77), and the intermediate patterns were defined between the short and persistent patterns.

## Extraction of virus–prokaryote pairs related to prokaryotic production in the surface layer

To extract virus–prokaryote infection pairs that could contribute to the prokaryotic production in the surface layer during the summer (middle of the stratified period), when bacterial production is high (July–September 2019, Fig. 1C), we performed co-occurrence network analysis with $ASV_{dominant}$ and LBVs. The LBVs those hosts were successfully predicted (phylum and class level) were applied to the co-occurrence network analysis. The $ASV_{dominant}$ was defined as the trend of relative abundance that was the highest in the middle of the stratified period in 2019 and when the highest relative abundance was more than 1%. Infectious pairs between $ASV_{dominant}$ and LBVs were successfully predicted and extracted using extended local similarity analysis (78, 79) via Pearson's correlation test ($P < 0.05$, Q < 0.05). To determine if LBV increases significantly as putative hosts ($ASV_{dominant}$) increases, we compared LBV abundance between months during which the putative host ($ASV_{dominant}$) was at least 1% during high prokaryotic production in 2019 and other months in 2019 outside of the high prokaryotic production period.

## Data processing and visualization

Data handling was performed using the tidyverse v.1.3.1 package (80), and all figures were visualized using the ggplot2 v. 3.3.5 package (81) in R language. For the samples that met normality (Kolmogorov–Smirnov test) and equal variance assumptions (F-test), as well as those that met normality but not the equal variance assumption, the Student's $t$-test and Welch's $t$-test were used, respectively. The Mann–Whitney U-test was applied to samples that did not meet these assumptions. Based on these thresholds, the Student's and Welch's $t$-tests were applied to test the statistical differences in prokaryotic and viral abundance, and prokaryotic production between the surface and deep layer (Table S1). The Mann–Whitney U-test and Student's $t$-test were applied to test the statistical significance of the observed differences in the dissimilarity of

time-series separation (Table S3). The Mann–Whitney U-test was applied to test the statistical significance of the increase in LBV abundance with an increase in the putative host $ASV_{dominant}$ (Table S7). Results with $P < 0.05$ were considered statistically significant.

## ACKNOWLEDGMENTS

We thank the Lake Biwa Environmental Research Institute for assistance with field sampling. We also thank SuperComputer System, Institute for Chemical Research, Kyoto University for providing computer time. This research was supported by JSPS KAKENHI (grant numbers JP19J14985, JP20H04323, JP20K12140, JP21H05057, JP22K14351, and JP22J01607), Kurita Water and Environment Foundation (19B045), and Kyoto University Foundation.

## AUTHOR AFFILIATIONS

[1]Research Center for Environmental Quality Management, Kyoto University, Otsu, Shiga, Japan
[2]Lake Biwa Branch Office, National Institute for Environmental Studies, Otsu, Shiga, Japan
[3]Department of Civil and Environmental Engineering, Ritsumeikan University, Kusatsu, Japan
[4]Graduate School of Frontier Sciences, The University of Tokyo, Kashiwa, Chiba, Japan
[5]Regional Environment Conservation Division, National Institute for Environmental Studies, Tsukuba, Ibaraki, Japan
[6]Graduate School of Agriculture, Kyoto University, Kitashirakawa-Oiwake, Sakyo-ku, Kyoto, Japan

## AUTHOR ORCIDs

Shang Shen  http://orcid.org/0000-0002-4293-8967
Kento Tominaga  http://orcid.org/0000-0003-4814-5690

## FUNDING

| Funder | Grant(s) | Author(s) |
| --- | --- | --- |
| MEXT | Japan Society for the Promotion of Science (JSPS) | 19J14985, 22K14351, 22J01607 | Shang Shen |
| MEXT | Japan Society for the Promotion of Science (JSPS) | 20H04323 | Yoshihisa Shimizu |
| MEXT | Japan Society for the Promotion of Science (JSPS) | 20K12140 | Kenji Tsuchiya |
| MEXT | Japan Society for the Promotion of Science (JSPS) | 21H05057 | Takashi Yoshida |
| Kurita Water and Environment Foundation (KWEF) | 19B045 | Shang Shen |

## AUTHOR CONTRIBUTIONS

Shang Shen, Conceptualization, Data curation, Formal analysis, Funding acquisition, Investigation, Methodology, Writing – original draft | Kento Tominaga, Methodology, Software, Validation, Writing – review and editing | Kenji Tsuchiya, Formal analysis, Investigation, Methodology, Writing – review and editing | Tomonari Matsuda, Conceptualization, Methodology, Validation, Writing – review and editing | Takashi Yoshida, Methodology, Resources, Validation, Writing – review and editing | Yoshihisa Shimizu, Funding acquisition, Investigation, Writing – review and editing

## DATA AVAILABILITY

Raw reads of the 16S rRNA genes from prokaryotic fractions have been deposited in GenBank (SAMD00576609-SAMD00576640). Raw reads of the viral fractions have been deposited in GenBank (SAMD00576528-SAMD00576554).

## ADDITIONAL FILES

The following material is available online.

### Supplemental Material

**Supplemental Figures (mSystems00906-23-s0001.docx).** Figures S1-S5.
**Supplemental Tables (mSystems00906-23-s0002.xlsx).** Tables S1-S11.

### Open Peer Review

**PEER REVIEW HISTORY**
**(review-history.pdf).** An accounting of the reviewer comments and feedback.

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
