## [Reviewer comments · mSystems]

Virus–prokaryote infection pairs associated with prokaryotic production in a freshwater lake

Shang Shen, Kento Tominaga, Kenji Tsuchiya, Tomonari Matsuda, Takashi Yoshida, and Yoshihisa Shimizu

Corresponding Author(s): Shang Shen, Ritsumeikan Daigaku College of Science and Engineering

Review Timeline:

Submission Date:	August 25, 2023
Editorial Decision:	October 4, 2023
Revision Received:	October 12, 2023
Editorial Decision:	October 31, 2023
Revision Received:	November 3, 2023
Editorial Decision:	November 7, 2023
Revision Received:	November 8, 2023
Accepted:	December 6, 2023

Editor: Ryan McClure

Reviewer(s): Disclosure of reviewer identity is with reference to reviewer comments included in decision letter(s). The following individuals involved in review of your submission have agreed to reveal their identity: Roberto Vázquez (Reviewer #2)

Transaction Report:

DOI: <https://doi.org/10.1128/msystems.00906-23>

October 4, 2023

Dr. Shang Shen
Ritsumeikan Daigaku College of Science and Engineering
Kusatsu, Shiga 525-8577
Japan

Re: mSystems00906-23 (**Virus-prokaryote infection pairs associated with prokaryotic production in a freshwater lake**)

Dear Dr. Shang Shen:

Thank you for submitting your manuscript to mSystems. The reviewers were pleased with much of your changes but one reviewer still has one major concern. Therefore, acceptance will not be final until you have adequately addressed the reviewer comments.

Preparing Revision Guidelines

Please return the manuscript within 60 days; if you cannot complete the modification within this time period, please contact me. If you do not wish to modify the manuscript and prefer to submit it to another journal, please notify me of your decision immediately so that the manuscript may be formally withdrawn from consideration by mSystems.

Sincerely,

Ryan McClure

Editor, mSystems

Journals Department
Reviewer comments:

Reviewer #2 (Comments for the Author):

As far as my expertise goes, I do not have any more issues to discuss. I would like to thank the authors for taking the time to address the issues raised in the previous review round in such depth.

Just one final minor comment from my side:

In Table S1 you say original data are shown in Figure S6 which, as far as I can tell does not exist. It rather refers to Figure 1CDE if I am understanding it correctly. Authors should also explicitly clarify that what's being compared in the statistical analysis is surface vs deep layer (again, if I am getting it right).

Overview: The authors have responded to most of my suggestions, and I am mostly satisfied with their responses. I think that the manuscript reads well and their effort in revision is appreciated. I recommend further clarification of the statistical methods, including specific tests used, which should be mentioned throughout the manuscript and in associated figure legends. I also note that one of my more major comments was not addressed (viral identification reanalysis/comparison; see below), which I leave up to the editor to adjudicate.

Minor comments:

Lines 367-369: “Although...”

I don't fully agree with this sentiment as the reason that we think most viruses are dsDNA viruses is because of a historical attention focused on dsDNA viruses (partially due to technological and methodological limitations). I would recommend phrasing that reflects this, such as “Although ssDNA and RNA viruses infect prokaryotes, most of those described prokaryote-infecting viruses are dsDNA viruses, and as such we have focused our attention on this group.” I would recommend a similar approach in your Discussion paragraph about this (lines 247-255), but I do appreciate the efforts made by the authors to address this throughout the manuscript, as well as associated limitations.

Lines 367-369: “raw reads were assembled using SPAdes ...”

Thank you for clarifying that you used default SPAdes. I will say that I find it curious that you chose to use SPAdes in its default format as opposed to metaspades or single cell spades, which is preferred for metagenomic data. I suspect that this would likely result in more misassemblies and merging of contigs from unrelated taxa (as default spades assumes the input is from an isolate genome), but given that the rest of your methods are fairly conservative, my guess/hope is that the effect of any associated misassembly would be mostly mitigated? Just a guess. I would recommend using single cell SPAdes in your future work, as Simon Roux put out a good paper a few years back (<https://peerj.com/articles/6902/>) examining the effect of different assembly approaches and found scSPAdes to generate the best assemblies and this has been my experience as well.

Major comments:

Response to “As suggested, these tools can detect more viral contigs than Virsorter1, which we used in this study...”

This isn't really a sufficient response. Yes, you found viruses associated with prokaryotic production, but there's no ground truth that says you found all/most of them. In fact, it is almost certain that most other methods would more accurately detect viral contigs, and, importantly help with accurate annotation in light of potential misassembly. Virsorter1 is an antiquated tool that is outperformed by all other methods, which may be of particular importance given your usage of default SPAdes. I will leave it up to the editor to decide

how to respond here, but I do not find this to be a satisfactory response. I think a comparison with another tool should be made or a contemporary tool like geNomad should be used if a comparison is not desired.

Please expand on the statistical analyses used. This aspect of the study is far too scant. There is a brief section under the data processing, but the authors should specify the specific tests used for each set of analyses, (eg. for ecological analyses – permanova?) and how multiple hypothesis correction was performed. This is still very underdeveloped and needs expanded upon pretty significantly throughout the manuscript. Please also list the units and tests used for each analysis in the associated figure legends (the supp figs would be difficult for readers to interpret if they were not also looking at the manuscript text). I found myself often looking at figures/reading text and wondering what tests were used and if the results were significantly different or just passed the eyeball test.

Reviewer 1

Overview: The authors have responded to most of my suggestions, and I am mostly satisfied with their responses. I think that the manuscript reads well and their effort in revision is appreciated. I recommend further clarification of the statistical methods, including specific tests used, which should be mentioned throughout the manuscript and in associated figure legends. I also note that one of my more major comments was not addressed (viral identification reanalysis/comparison; see below), which I leave up to the editor to adjudicate.

Minor comments:

Lines 367-369: “Although...”

I don't fully agree with this sentiment as the reason that we think most viruses are dsDNA viruses is because of a historical attention focused on dsDNA viruses (partially due to technological and methodological limitations). I would recommend phrasing that reflects this, such as “Although ssDNA and RNA viruses infect prokaryotes, most of those described prokaryote-infecting viruses are dsDNA viruses, and as such we have focused our attention on this group.” I would recommend a similar approach in your Discussion paragraph about this (lines 247-255), but I do appreciate the efforts made by the authors to address this throughout the manuscript, as well as associated limitations.

Response:

Thank you for this suggestion. We have revised the relevant parts accordingly (Lines 250, 367–369).

Lines 367-369: “raw reads were assembled using SPAdes ...”

Thank you for clarifying that you used default SPAdes. I will say that I find it curious that you chose to use SPAdes in its default format as opposed to metaspades or single cell spades, which is preferred for metagenomic data. I suspect that this would likely result in more misassemblies and merging of contigs from unrelated taxa (as default spades assumes the input is from an isolate genome), but given that the rest of your methods are fairly conservative, my guess/hope is that the effect of any associated misassembly would be mostly mitigated? Just a guess. I would recommend using single cell SPAdes in your future work, as Simon Roux put out a good paper a few years back (<https://peerj.com/articles/6902/>) examining the effect of different assembly approaches and found scSPAdes to generate the best assemblies and this has been my experience as well.

Response:

Thank you for your comment and for recommending a good paper for reference. Misassembly was carefully addressed through bioinformatics and PCR amplification experiments when establishing this method in the below paper. Finally we chose SPAdes with “careful” flags based on a previous work in which several assemble tools were compared and SPAdes was identified as the best assembler (Table 1 in Nishimura et al., 2017). Collinearity with the isolated viruses was observed in the assembled genomes. And although there were surprisingly few SNPs, the sequences were confirmed by PCR amplification of some parts with relatively low coverage from environmental samples. Thus, we conclude that the genomes assembled using the method is not derived from a single virus particle, but is a consensus genome among closely related viruses as long as they exist in that environment (Nishimura et al., 2017). Per your suggestion, we aim to use scSPAdes (or metaSPAdes) as an assembler in the subsequent viral metagenome investigations.

Nishimura, Y., Watai, H., Honda, T., Mihara, T., Omae, K., Roux, S., Blanc-Mathieu, R., Yamamoto, K., Hingamp, P., Sako, Y., Sullivan, M. B., Goto, S., Ogata, H., & Yoshida, T. (2017). Environmental viral genomes shed new light on virus-host interactions in the ocean. *MSphere*, 2(2), e00359-16. <https://doi.org/10.1128/mSphere.00359-16>

Major comments:

Response to “As suggested, these tools can detect more viral contigs than Virsorter1, which we used in this study...”

This isn't really a sufficient response. Yes, you found viruses associated with prokaryotic production, but there's no ground truth that says you found all/most of them. In fact, it is almost certain that most other methods would more accurately detect viral contigs, and, importantly help with accurate annotation in light of potential misassembly. Virsorter1 is an antiquated tool that is outperformed by all other methods, which may be of particular importance given your usage of default SPAdes. I will leave it up to the editor to decide how to respond here, but I do not find this to be a satisfactory response. I think a comparison with another tool should be made or a contemporary tool like geNomad should be used if a comparison is not desired.

Response:

Thank you for this comment. We agree that most of the other tools are more accurate than VirSorter 1. However, we wanted to emphasize that we purified the viral particles using CsCl step-gradient ultracentrifugation and DNase I steps, by which the remaining prokaryotic cells and external DNA were removed before bioinformatic analysis. This method was originally used for the purification of lambda phages (dsDNA) and has

occasionally been used to purify viruses before virome analysis. Indeed, in our study, 86% of the assembled contigs (>10 kb) were detected to be of viral origin by VirSorter1 (Table S4). In contrast, a previous study that did not use a CsCl step-gradient ultracentrifugation step detected only half of the contigs to be of viral origin using VirSorter1 (EX. 2 in Luo et al., 2020). These observations indicate that our method (CsCl step-gradient ultracentrifugation + SPAdes + VirSorter1) worked well and could sufficiently capture a viral population. We believe that our method is sufficient to achieve our goal, such that we need not perform a reanalysis or comparison with other methods. In the method text, virus concentration and purification steps were described separately (Lines 366-369). The GeNomad paper was published on September 21, 2023, and our manuscript has been under review during this period. We will consider this tool for our subsequent virome analyses.

Luo, E., Eppley, J.M., Romano, A.E. *et al.* Double-stranded DNA viroplankton dynamics and reproductive strategies in the oligotrophic open ocean water column. *ISME J* **14**, 1304–1315 (2020). <https://doi.org/10.1038/s41396-020-0604-8>

Please expand on the statistical analyses used. This aspect of the study is far too scant. There is a brief section under the data processing, but the authors should specify the specific tests used for each set of analyses, (eg. for ecological analyses – permanova?) and how multiple hypothesis correction was performed. This is still very underdeveloped and needs expanded upon pretty significantly throughout the manuscript. Please also list the units and tests used for each analysis in the associated figure legends (the supp figs would be difficult for readers to interpret if they were not also looking at the manuscript text). I found myself often looking at figures/reading text and wondering what tests were used and if the results were significantly different or just passed the eyeball test.

Response:

Thank you for your comment. We have revised the Data Processing and Visualization section to include more details (Lines 451–460). We have also added units and tests used for each analysis in the figure captions (Lines 734–736, 746–748, caption of Figure S5). We have also added the results of the test in the Result section (Lines 116, 121, 139, 160).

Reviewer 2

As far as my expertise goes, I do not have any more issues to discuss. I would like to thank the authors for taking the time to address the issues raised in the previous review

round in such depth.

Just one final minor comment from my side:

In Table S1 you say original data are shown in Figure S6 which, as far as I can tell does not exist. It rather refers to Figure 1CDE if I am understanding it correctly. Authors should also explicitly clarify that what's being compared in the statistical analysis is surface vs deep layer (again, if I am getting it right).

Response:

Thank you for your comment. We have revised the caption (added “between the surface and deep layer”) and the figure number (Figure 1C–E) (Table S1) accordingly.

Re: mSystems00906-23R1 (**Virus-prokaryote infection pairs associated with prokaryotic production in a freshwater lake**)

Dear Dr. Shang Shen:

Thank you for the privilege of reviewing your work. Below you will find my comments, instructions from the mSystems editorial office, and the reviewer comments. In principle I believe we can accept the paper but there are a couple minor points from one of the reviewers below that need to be addressed before we can do that.

Revision Guidelines

Sincerely,
Ryan McClure
Editor
mSystems

Reviewer #2 (Comments for the Author):

I have nothing really substantial to add to the current point of debate in the review process.

I could, however, try to disentangle the issue beyond discussions about using one method or the other, in which I lack the

expertise to intervene.

According to reviewer#1, the authors "found viruses associated with prokaryotic production, but there's no ground truth that says [they] found all/most of them". Now if I am understanding correctly, this is the key issue at stake. I may even agree that it cannot be claimed that most or all viral diversity was detected. Assuming this, then my questions are:

(1) is it implied by the authors in the manuscript that they found most/all viral diversity? If so, it may be worth it to nuance the text a bit.

(2) does the fact that not all viral diversity was captured substantially affect the authors claims?

Reviewer #2 (Comments for the Author):

I have nothing really substantial to add to the current point of debate in the review process. I could, however, try to disentangle the issue beyond discussions about using one method or the other, in which I lack the expertise to intervene.

According to reviewer#1, the authors "found viruses associated with prokaryotic production, but there's no ground truth that says [they] found all/most of them". Now if I am understanding correctly, this is the key issue at stake. I may even agree that it cannot be claimed that most or all viral diversity was detected. Assuming this, then my questions are:

- (1) Is it implied by the authors in the manuscript that they found most/all viral diversity? If so, it may be worth it to nuance the text a bit.

Response: No, we did not imply in the manuscript that we found all or most viral diversity. As you and Reviewer 1 have commented, not all viruses were captured in this study. The 13,761 viral contigs detected in this study were combined with approximately 4,000 viral contigs recovered from other 12 metagenomic samples collected from the same lake (Okazaki et al., 2019: <https://doi.org/10.1111/1462-2920.14816>), and all contigs were clustered. In total, 14,952 viral contigs were obtained (Figure below). It seems to be becoming saturated, but some viruses could still be detected. Thus we want to correct our response in the previous round. We have responded that “our method (CsCl step-gradient ultracentrifugation + SPAdes + VirSorter1) worked well and could sufficiently capture a viral population” and want to replace with “our method worked well and VirSorter 1 could sufficiently detect viruses from assembled contigs”. Thank you for your comment.

(2) Does the fact that not all viral diversity was captured substantially affect authors' claims?

Response: No, we believe that this will not significantly affect our claims. Our goal was to extract virus–prokaryotic infection pairs associated with prokaryotic production. Using 13,761 viral contigs, we successfully extracted co-occurring viruses for every prokaryote associated with prokaryotic production. This indicates that our goal had already been achieved. If we could capture all viruses, more co-occurring viruses could be extracted, but this would not affect our results.

Re: mSystems00906-23R2 (**Virus-prokaryote infection pairs associated with prokaryotic production in a freshwater lake**)

Dear Dr. Shang Shen:

After reviewing the comments from Reviewer 2 it seems as though you have addressed everything except for one final issue. Reviewer 2 had a question about your choice of assembly protocol and viral identification and annotation methods. It seems that there have been other programs published recently. Can you add a few lines in the Results or Methods as to why you chose these particular methods? I believe if you can justify the choice then we will be able to publish the paper. Please return the manuscript within 60 days; if you cannot complete the modification within this time period, please contact me. If you do not wish to modify the manuscript and prefer to submit it to another journal, notify me immediately so that the manuscript may be formally withdrawn from consideration by mSystems.

Revision Guidelines

Sincerely,
Ryan McClure
Editor
mSystems

After reviewing the comments from Reviewer 2 it seems as though you have addressed everything except for one final issue. Reviewer 2 had a question about your choice of assembly protocol and viral identification and annotation methods. It seems that there have been other programs published recently. Can you add a few lines in the Results or Methods as to why you chose these particular methods? I believe if you can justify the choice then we will be able to publish the paper.

Thank you for your comments. We have added some sentences/phrases to explain why we have chose our methods (Lines 380-381, 385-386, 391-392).

Re: mSystems00906-23R3 (**Virus-prokaryote infection pairs associated with prokaryotic production in a freshwater lake**)

Dear Dr. Shang Shen:

Your manuscript has been accepted, and I am forwarding it to the ASM production staff for publication. Your paper will first be checked to make sure all elements meet the technical requirements. ASM staff will contact you if anything needs to be revised before copyediting and production can begin. Otherwise, you will be notified when your proofs are ready to be viewed.

Featured Image Submissions: If you would like to submit a potential Featured Image, please email a file and a short legend to mSystems@asmusa.org. Please note that we can only consider images that (i) the authors created or own and (ii) have not been previously published. By submitting, you agree that the image can be used under the same terms as the published article. File requirements: square dimensions (4" x 4"), 300 dpi resolution, RGB colorspace, TIF file format.

Sincerely,
Ryan McClure
Editor
mSystems